# Scaling Web Agent Training through Automatic Data Generation and Fine-grained Evaluation

**Lajanugen Logeswaran, Jaekyeom Kim, Sungryull Sohn, Creighton Glasscock, Honglak Lee**
{llajan,jaekyeom,srsohn,creiglas,honglak}@lgresearch.ai
LG AI Research

## Abstract

We present a scalable pipeline for automatically generating high-quality training data for web agents. In particular, a major challenge in identifying high-quality training instances is trajectory evaluation - quantifying how much progress was made towards task completion. We introduce a novel constraint-based evaluation framework that provides fine-grained assessment of progress towards task completion. This enables us to leverage partially successful trajectories, which significantly expands the amount of usable training data. We evaluate our method on a new benchmark we propose called *BookingArena*, which consists of complex booking tasks across 20 popular websites, and demonstrate that our distilled student model outperforms open-source approaches and matches or exceeds commercial systems, while being a significantly smaller model. Our work addresses the challenge of efficiently creating diverse, realistic web interaction datasets and provides a systematic evaluation methodology for complex structured web tasks.

## 1 Introduction

The rise of large language models (LLMs) has spurred significant interest in web agents—systems capable of navigating and interacting with complex, dynamic websites to accomplish real-world tasks. These agents face substantial challenges: each web page can contain hundreds of interactive elements, and successful task completion requires making sequential decisions across multiple pages.

Advances in agents capable of fluently performing tasks in virtual environments such as the web rely on two critical, interrelated ingredients: data and automatic evaluation. Large-scale agent trajectory data is essential for training competent models. Due to the scale required, synthetic data generation becomes crucial. However, this introduces the challenge of verifying that the generated samples are meaningful and of high quality. The sequential decision-making nature of the tasks, combined with the inherent complexity of the web environment, makes this a particularly difficult problem.

In this work, we aim to address both challenges by proposing an automatic data generation pipeline, along with novel automatic evaluation metrics capable of assessing agent trajectories with high fidelity. Our approach focuses on scalable trajectory data generation—a relatively underexplored but crucial area for web agent development. We propose a fully automatic data generation pipeline that leverages publicly available large language models to produce large scale synthetic trajectories. Our approach targets complex, real-world web navigation tasks that involve multiple constraints such as travel booking scenarios.

The complexity of these multi-constraint tasks renders traditional evaluation approaches insufficient. Existing trajectory evaluation frameworks that rely on vision-language models (VLMs) or large language models (LLMs) as judges (He et al., 2024; Pan et al., 2024a; Trabucco et al., 2025; Pahuja et al., 2025) primarily depend on the models' ability to interpret requirements described in natural language. This reliance limits their effectiveness when applied to complex tasks with nuanced or multi-faceted constraints. To overcome

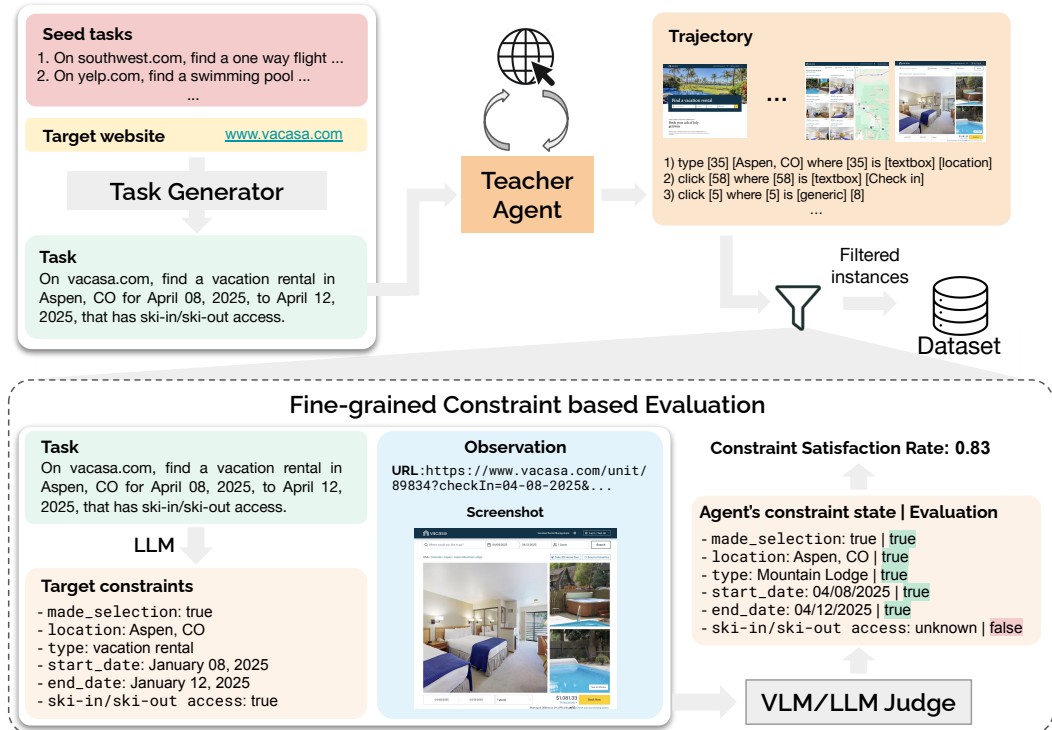

Figure 1: **Overview**. We introduce a scalable pipeline to automatically generate and evaluate web agent trajectories for training competent small language models (Section 2.1). Given trajectories generated by a few-shot prompted teacher agent, our novel constraint-based fine-grained evaluation framework (Section 2.2) extracts high-quality training instances for distillation. See text for details.

this limitation, we propose a novel constraint-based evaluation framework that enables fine-grained assessment of agent trajectories. Specifically, this framework examines each individual criterion specified in the task description and verifies whether it is satisfied in the resulting trajectory. Compared to binary success/failure metrics, our evaluation approach offers a more nuanced and interpretable measure of task progress while remaining robust to the complexities introduced by multi-constraint tasks.

Leveraging these innovations, we curate a large high quality training dataset. Our 24B parameter model finetuned on this data outperforms prior open-source as well as commercial systems, demonstrating the effectiveness of our overall pipeline. We make the following main contributions in this work:

- We propose a scalable automatic data generation pipeline which leverages few-shot prompted publicly available language models to synthesize large number of agent trajectories.

- We propose a novel constraint-based evaluation framework which enables fine-grained assessment and filtering of trajectories, enabling the curation of a high quality training dataset for agent training.

- We introduce *BookingArena*, a new benchmark consisting of 120 complex and structured tasks on real-world booking websites.

- Evaluation on complex booking tasks across 20 popular websites reveals that our distilled 24B parameter model outperforms both open-source alternatives and matches or exceeds the performance of commercial systems. Notably, our student model achieves better results than the much larger 405B parameter teacher model used for trajectory generation.

## 2 Approach

We present a scalable pipeline for generating high-quality data for training competent web agents. We first present our trajectory generation and distillation pipeline for generating large-scale web agent trajectory data from the web with few-shot prompted large language model agents in Section 2.1. Next, we propose a new fine-grained constraint-based evaluation framework for automatic evaluation and curation of trajectories in Section 2.2. See Figure 1 for an overview of our pipeline.

### 2.1 Automatic Trajectory Generation and Distillation

Our data generation pipeline consists of four key components described next: Task Generation, Trajectory Generation, Trajectory Evaluation and Distillation.

**Task Generation.** We begin by compiling a list of the 1,000 most popular safe websites, removing any that are inaccessible (e.g., those returning error codes). We then prompt GPT-4 with manually curated seed tasks to generate a diverse set of target tasks, following a few-shot learning paradigm (See Figure 3 of the Appendix for the prompt). The prompt encourages the model to generate tasks that are diverse, realistic, and span varying difficulty levels.

**Trajectory Generation.** Given a set of tasks, we employ a few-shot prompted Large Language Model (LLM) agent to generate trajectories by interacting with the web. The agent's prompt template (shown in Figure 4 of the Appendix) is adapted from Shi et al. (2024). At each timestep, the agent receives as input the task description, current URL, page's accessibility tree, history of previous actions and reasoning, and currently open browser tabs. The accessibility tree provides a structured representation of the webpage, where each element is formatted as `[id] [tagType] [text content] [properties]`. Here, `id` is a unique numerical identifier, `tagType` specifies the HTML element type (e.g., button, link), `text content` contains the element's visible text, and `properties` lists any additional attributes. Based on these inputs, the agent predicts the next action $a_t$ from a set of possible operations including element clicks, text input, hover events, and page navigation actions. The agent also produces a (chain-of-thought) reasoning for the predicted action. A trajectory $\tau$ is defined as sequence of action-observation pairs $\{(a_t, o_t)\}_{t=1}^{T}$. For each observation, we log metadata including the URL, accessibility tree, and page screenshot. This stage produces a collection of task description $d_i$ and corresponding agent trajectory $\tau_i$ pairs $\mathcal{D} = \{(d_i, \tau_i)\}_{i=1}^{N}$.

**Trajectory Evaluation.** Trajectories generated by the few-shot agent inevitably fail due to various factors including the limited capabilities of the underlying LLM, impossible tasks, and bot detection mechanisms. Therefore, a reliable and efficient automatic evaluation framework is essential for identifying high-quality trajectories suitable for dataset construction. Evaluating web agents presents significant challenges, particularly for tasks involving multiple constraints that must be satisfied. Prior approaches to automatic evaluation largely adopt an LLM-as-judge (Or Vision Language Model (VLM) as judge) paradigm where an LLM/VLM is asked to directly judge success/failure given screenshots from an agent trajectory (He et al., 2024). While effective for simple, short-horizon tasks, the scalability of this approach to more complex, multi-criteria tasks remains unclear. One of our key contributions is a comprehensive evaluation framework to automatically evaluate agent trajectories (Section 2.2).

**Distillation.** Using our automatic evaluation framework, we curate a high-quality training set $\mathcal{D}^{\text{train}}$ for distillation. The student model undergoes supervised fine-tuning to mimic the teacher model's behavior, learning to predict both the next action and its accompanying reasoning given the current observation and agent history.

## 2.2 Constraint Framework

We propose a fine-grained metric to evaluate task completion. Figure 1 illustrates our constraint-based evaluation process. Given a task, we first identify a set of constraints that must be satisfied in order to complete the task using an LLM. For instance, consider the task "Find a hotel in Paris for the dates Aug 2 - 3, 2025". An agent that successfully completes this task must fulfill the following constraints: {location: *Paris*, start_date: *Aug 2, 2025*, end_date: *Aug 3, 2025*}. Our evaluation metric examines how many of these individual constraints the agent satisfied during task execution, providing a granular assessment of performance. Traditional binary success/failure metrics treat all failures equally, offering no distinction between an agent that made no progress toward the task and one that made substantial progress but failed to fulfill all requirements. In contrast, constraint-based evaluation provides a more nuanced signal that can differentiate between these scenarios, offering valuable insights into partial task completion. This approach also enhances evaluation objectivity by introducing a systematic framework that reduces subjective judgment in performance assessment.

**Automatic Evaluation based on Constraints.** Having defined the relevant constraints for each task, we define an evaluation metric termed *Constraint Satisfaction Rate (CSR)*. CSR computes a binary success/failure score for each constraint representing whether the agent satisfied the corresponding constraint in task execution and computes the average score across all constraints. This metric is applicable to both LLMs and VLMs judges. For a VLM judge, we use the task description, final web page screenshot as well as the URL for evaluating constraint satisfaction. For an LLM judge, we use the final web page accessibility tree instead of the screenshot. Figure 5 shows the prompt used for constraint-based automatic evaluation.

Given a task $d$, from which a corresponding set of constraints $C = \{(c_i, v_i)\}_{i=1}^n$ are identified (e.g., using an LLM), we define the *Constraint Satisfaction Rate (CSR)* for a given observation $o$ as follows:

$$\text{CSR}_d(o) = \frac{1}{n} \sum_{i=1}^n \mathbb{I}\left[f(c_i|o,d) = v_i\right] \tag{1}$$

where $f$ represents an LLM/VLM judge that predicts the observed value of the constraint $c_i$ given the observation $o$ and task description $d$. Given a trajectory $\tau = \{(a_t, o_t)\}_{t=1}^T$, the constraint satisfaction rate is defined as the CSR of the final observation: $\text{CSR}_d(\tau) = \text{CSR}_d(o_T)$. The task success rate is defined as

$$\text{SR}_d(\tau) = \mathbb{I}\left[\text{CSR}_d(\tau) = 1\right] \tag{2}$$

These metrics are extended to a collection of trajectories by macro-averaging.

**Dataset Curation with Constraint-based Evaluation.** In addition to providing a fine-grained evaluation metric, constraints can also be helpful in evaluating progress towards the task. This provides an opportunity to leverage partial trajectories for training. This is important since teacher agents' success rates on hard tasks will be low and restricting to successful trajectories alone will significantly limit the amount of training data. While task success/failure metrics are only meaningful at the end of task completion, constraint satisfaction can provide a meaningful evaluation for partial trajectories as well. At any stage during task execution, CSR can be evaluated to assess the number of constraints fulfilled up to that point (illustrated in Figure 2).

Given a task description $d$ and trajectory $\tau = \{(a_t, o_t)\}_{t=1}^T$, we examine the maximum CSR value achieved at any point during the trajectory ($C_{\max} = \max_t \text{CSR}_d(o_t)$). If $C > 0$, we extract the trajectory prefix $\tau' = \{(a_t, o_t)\}_{t=1}^{t'}$ where $t'$ is the smallest time-step for which $\text{CSR}_d(o_{t'}) = C_{\max}$ (i.e., $t' = \min_t\{t : \text{CSR}_d(o_t) = C_{\max}\}$).

In the trajectory prefix extraction process described above, note that invalid 'stop' actions (i.e., action that marks the end of task execution) could have been included. These are actions $a_t = \text{stop}$ where $\text{CSR}_d(o_t) \neq 1$ – instances where agent prematurely predicted a stop

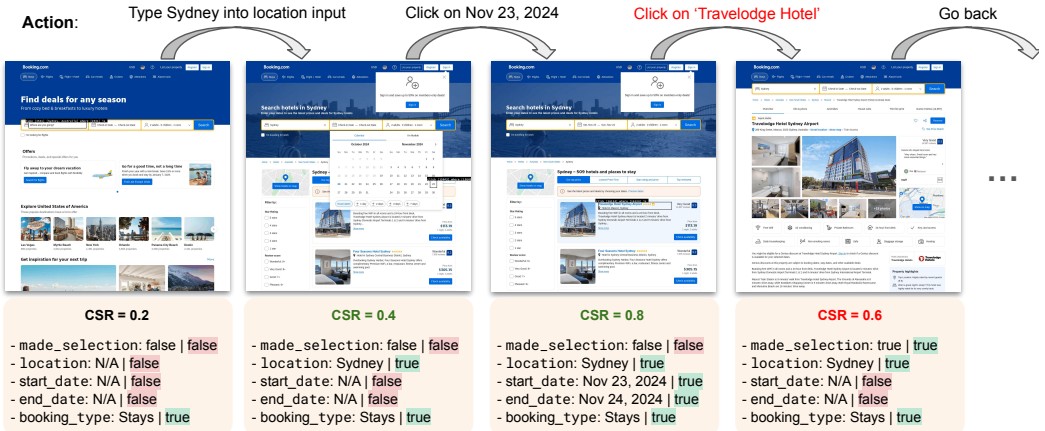

Task: On booking.com, find a hotel in Sydney, Australia for one night on November 23, 2024.

Figure 2: **Prefix Extraction with Constraint Evaluation**: For a given trajectory, we compute the constraint satisfaction rate (CSR) for each time-step and extract the smallest prefix that reaches maximum CSR. In this example, only the first two actions are retained and the third action is discarded as it results in a decrease of CSR (Agent should have clicked on the 'search' button instead).

action despite not all constraints of the task being met. We retain a stop action $a_t = \text{stop}$ if and only if $\text{CSR}_d(o_t) = 1$. For stop actions with $\text{CSR}_d(o_t) < 1$, we use a hindsight re-labeling approach to revise these for inclusion in the dataset described next.

**Hindsight Re-labeling.** For stop actions where the agent made partial progress toward completing the task, $0 < \text{CSR}_d(o_t) < 1$, we rename the task description so that these become valid stop actions with the following hindsight re-labeling approach. Given that $C' \subset C$ constraints were satisfied, we revise the original task description $d$ to $d'$ so that the new description is consistent with the constraints $C'$ (i.e., constraints $C \setminus C'$ are removed from the task description). For example, consider the task "On hilton.com, find a hotel in New Orleans on February 08 for 3 people". Suppose the status of constraints after task execution was judged to be { made_selection: *False*, location: *"New Orleans"*, date: *"February 8, 2025"*, guest_count: *<Unobserved>* }. After hindsight re-labeling, the task is revised to "On hilton.com, search for hotels in New Orleans on February 8". In addition, we also revise the agent's reasoning steps for the stop action to be consistent with the new task description.

## 3 Experiments

**Dataset Construction.** We use LLAMA 3 70B and LLAMA 3.1 405B Instruct models (Touvron et al., 2023) as teacher agents for trajectory generation, where the bigger model was necessary for complex tasks (e.g, booking). We collect 150k trajectories, which correspond to approximately 1M individual actions/steps (note that this includes successful and unsuccessful trajectories). The generated trajectories are evaluated with our constraint evaluation approach using LLAMA 3.3 70B and Gemma 3 27B (Team et al., 2025) models (Prompt provided in Figure 6 of Appendix). These evaluation outputs are used by the prefix extraction and hindsight relabeling steps (Section 2.2) for curating a high-quality training dataset. To counter evaluation noise, we only retain training instances that are common to both these judge models. From the 150k trajectories collected, ~16k were judged to be completely successful (based on our constraint evaluation metric). On the other hand, if we consider trajectories that are partially correct (i.e., CSR > 0 for at least one step), we find that ~65k trajectories have usable prefixes, which leads to a dataset of 300k actions/steps. All experiments are conducted using two machines, each equipped with 16 A100 40GB GPUs.

**Distillation.** We train Mistral 3 Small 24B (Team, 2025) student agents. The prompt used for fine-tuning is shown in Figure 8 of the Appendix. We use LoRA fine-tuning for distillation, where we fine-tune the query and value projections (q_proj, v_proj). We use a learning rate of 1e-4 with a cosine scheduler and a batch size of 16.

**Evaluation Setup.** In contrast to some prior work that use static datasets for evaluation, which have been shown to poorly correlate with task success rate in the wild (Zheng et al., 2024), we focus on online tasks, where the agent is expected to execute a task on a website and success/failure evaluation is performed at the end of trajectory. We consider multiple evaluation sets for accurate assessment of different agents' performance on web tasks.

Since existing benchmarks for online evaluation lack representation of complex long-horizon tasks, we develop a new benchmark termed **BookingArena** focused on travel booking tasks (flights, hotels, tickets, etc.). This benchmark comprises 6 tasks from each of 20 popular websites commonly used for real-world booking activities. We define difficulty levels based on the number of constraints specified in each task. The 6 tasks per website are distributed as 2 easy, 2 medium, and 2 hard tasks, with average number of constraints for each category being 3.95, 5.45, and 7.30 respectively. Each task is identified by a starting url and task description. Agents are required to interact with actual websites and complete tasks. To prevent tasks from being outdated with the passing of time, we also provide scripts that automatically detect past dates and update them with future dates to keep the tasks valid. We hope this benchmark will be useful to study complex, long-horizon tasks in the future.

In addition, we also evaluate our approach on **WebVoyager** (He et al., 2024), which is dominated by search tasks. This benchmark comprises 643 tasks from 15 websites, with an average of 43 tasks per website. 55 tasks that are no longer possible (e.g., due to their time-sensitive nature), are excluded from our evaluation. For booking tasks that are fixable by updating dates (e.g., 'Find deals for a vacation in Mexico in Dec 2024'), we update these dates so that they are in the future.

The rest of this section is organized as follows. We begin by presenting the evaluation on our BookingArena benchmark in Section 3.1. Next, we provide ablation studies on our pipeline's design choices (Section 3.2) and present a study on our automatic evaluation approach (Section 3.3). Finally, we present results on the WebVoyager benchmark in Section 3.4.

### 3.1 BookingArena Evaluation

**Baselines.** We consider commercial and open-source state-of-the-Art VLM agents as baselines: UI-TARS (Qin et al., 2025), Claude-Computer-Use (`computer-use-2024-10-22`) (Anthropic, 2024), Operator (`computer-use-preview-2025-03-11`) (OpenAI, 2025), Browser Use (Müller & Žunič, 2024). Browser Use is a set-of-marks-based agent where interactable screen elements are highlighted with colored bounding boxes with numerical ids and the agent predicts element ids to interact with. UI-TARS, Claude-Computer-Use, and Operator, on the other hand, are pixel-based approaches, which interact with screen elements based on pixel coordinates. These agents take one or more screenshots as input and produce actions.

**Evaluation Protocol.** We use the constraint-based success rates, SR and CSR, as defined in Equations (1) and (2) for this evaluation. For evaluation of the final models, we use GPT-4o with visual observation space (final screenshot and URL). The prompt used for evaluation is given in Figure 5 of the Appendix. Note that this is different from the models (LLAMA, Gemma) used to curate the dataset, alleviating concerns of dataset curation bias.

**Results.** Our approach outperforms prior methods based on closed-source models such as Claude Computer Use, Operator and Browser Use, as well as open-source alternatives such as UI-TARS on the task SR metric, and is only inferior to Operator. This is particularly significant, given that our agent is a 24B parameter model. However, we also find that SR is poor across the board due to the challenging nature of the tasks, with performance on several websites being zero. Task success rate is a stringent metric that equally penalizes an agent that got a majority of the constraints in the task correct, as well as an agent that

| Website | Browser Use | | Claude CU | | UI-TARS | | Operator | | Ours | |
|---|---|---|---|---|---|---|---|---|---|---|
| | SR | CSR | SR | CSR | SR | CSR | SR | CSR | SR | CSR |
| Booking.com | 20.0 | 44.0 | 0.0 | 6.7 | **66.7** | 83.3 | 0.0 | 55.6 | 50.0 | **89.2** |
| Vrbo | 0.0 | 40.3 | 33.3 | 62.0 | 0.0 | 26.9 | 16.7 | 68.5 | **66.7** | **88.4** |
| Trivago | 33.3 | 66.5 | 0.0 | 50.5 | 33.3 | 66.3 | **50.0** | 84.2 | 50.0 | **85.9** |
| Travelocity | 0.0 | 50.0 | 0.0 | 56.7 | **50.0** | 68.9 | 20.0 | 77.7 | 50.0 | **85.6** |
| Rome2rio | 40.0 | 71.7 | 33.3 | 58.3 | 0.0 | 25.0 | **50.0** | 75.0 | 50.0 | **81.9** |
| Orbitz | 16.7 | 53.1 | 33.3 | 49.6 | 16.7 | 28.9 | **83.3** | **98.3** | 80.0 | 80.0 |
| Homeaway | 16.7 | 40.5 | 16.7 | 59.5 | 33.3 | **76.8** | **50.0** | 66.7 | 50.0 | 73.8 |
| Airbnb | 0.0 | 9.7 | 0.0 | 34.1 | 0.0 | 56.8 | 20.0 | 66.0 | **33.3** | **72.9** |
| Kayak | **50.0** | 80.0 | 0.0 | 43.0 | 33.3 | 57.2 | 25.0 | **80.4** | 16.7 | 68.0 |
| Vacasa | 33.3 | 43.3 | 33.3 | 56.3 | 0.0 | 37.9 | **50.0** | **69.0** | 33.3 | 66.2 |
| Plumguide | **50.0** | **76.4** | 16.7 | 38.9 | 33.3 | 48.6 | 20.0 | 58.3 | 33.3 | 63.9 |
| IHG | 33.3 | 40.0 | 33.3 | 60.7 | 33.3 | 61.9 | **40.0** | **67.0** | 33.3 | 59.8 |
| Sonder | 0.0 | 10.0 | 16.7 | 30.0 | 0.0 | 11.7 | **25.0** | 56.7 | 16.7 | **56.7** |
| Redawning | 16.7 | 47.4 | 16.7 | 36.9 | **33.3** | **56.2** | 16.7 | 49.0 | 20.0 | 50.2 |
| Hotels.com | 16.7 | 62.2 | 16.7 | 59.2 | **66.7** | **80.0** | 25.0 | 65.0 | 16.7 | 48.3 |
| Momondo | 16.7 | 38.9 | 16.7 | 78.9 | 33.3 | 80.0 | **50.0** | **90.8** | 0.0 | 41.1 |
| Hilton | 16.7 | 36.1 | 50.0 | 76.7 | 50.0 | 84.4 | **80.0** | **96.0** | 0.0 | 33.3 |
| Google Flights | 0.0 | 27.5 | 16.7 | 60.5 | 33.3 | 59.2 | **40.0** | **88.1** | 0.0 | 31.2 |
| Radisson | **20.0** | 36.0 | 0.0 | 17.8 | 16.7 | 26.7 | 0.0 | **47.7** | 0.0 | 12.2 |
| Megabus | 0.0 | 34.4 | 0.0 | **36.1** | 0.0 | 35.0 | 0.0 | 23.9 | 0.0 | 12.2 |
| Average | 18.8 | 45.3 | 16.8 | 48.7 | 26.7 | 53.6 | **33.0** | **68.7** | 29.9 | 60.2 |

Table 1: Website-level performance comparison of different approaches on our BookingArena benchmark. We compare our approach against Browser Use (Müller & Žunič, 2024), Claude Computer Use (Anthropic, 2024), UI-TARS (Qin et al., 2025) and Operator (OpenAI, 2025). SR represents task success rate and CSR represents constraint satisfaction rate. Best performance boldfaced and second best underlined.

made no progress on a task. For such difficult tasks, we argue that CSR provides a more fine-grained signal to compare different models.

We find that Browser Use struggles with dense webpages, where the set-of-marks annotation can make the screenshot complex and difficult to parse. While Operator is generally capable with the best CSR among all approaches, we find that it often makes one-off errors with date selection (e.g., choosing 'May 14' when the provided date is 'May 15'), which explains the low SR. A key advantage of our proposed constraint-based metric is that it provides such fine-grained insights about the deficiencies of agents, which is critical for advancing the frontier on web agents.

We further evaluate how often a specific constraint is successfully satisfied by our agetnt across tasks. Success rates for the 10 most frequent constraints appearing in the evaluation tasks are shown in Table 3. Intuitively, location-related constraints have higher success rates, as they are often filled early in the task. In contrast, constraints related to amenities, such as pet-friendliness, are handled later in the task and have low success rates as a result, due to the dependency on prior actions.

## 3.2 Ablations on Agent Training

We perform ablations to understand the impact of different components of our approach such as the choice of teacher model, impact of our data filtering strategy, and fine-tuning design choices. For these ablations, we report SR and CSR on BookingArena in Table 2.

**Teacher Performance.** We compare different few-shot prompted teacher models with 3 demonstration examples in the first section of Table 2. We find that LLAMA models perform well, with the 405B model performing best on complex tasks.

| Model | Data Source | Training/ Inference | SR % | CSR % |
|---|---|---|---|---|
| Qwen 2.5 72B | N/A | Few-shot | 18.0 | 46.0 |
| Llama 3.3 70B | N/A | Few-shot | 22.5 | 49.8 |
| Llama 3.1 405B | N/A | Few-shot | 22.5 | 52.4 |
| Mistral 3 24B | Success only | LoRA | 25.0 | 55.0 |
| Mistral 3 24B | All trajectories | LoRA | 29.4 | 57.2 |
| Mistral 3 24B | Partial success | LoRA | **29.9** | **60.2** |
| Mistral 3 24B | Partial success | Full finetune | 28.4 | 58.7 |

Table 2: Performance comparison across model variants and prompting/training configurations. We evaluate teacher models using few-shot prompting and student models trained with different data filtering strategies and training approaches. Task SR and Constraint SR represent the task success rate and constraint satisfaction rate respectively.

| Constraint | Success Rate (%) |
|---|---|
| location | 79 |
| departure loc. | 73 |
| destination loc. | 67 |
| return date | 64 |
| departure date | 56 |
| start date | 55 |
| pet-friendly | 50 |
| end date | 47 |
| rental type | 41 |
| num guests | 25 |

Table 3: Success rates of our fine-tuned agent for the top-10 most frequent constraints in the evaluation tasks.

| Model | Task SR (%) | | | Constraint SR (CSR) (%) | | |
|---|---|---|---|---|---|---|
| | Screenshot | +URL | +History | Screenshot | +URL | +History |
| Claude Computer Use | 12.6 | 16.8 | 21.8 | 39.1 | 48.7 | 53.0 |
| LLAMA 405B few-shot | 19.0 | 22.5 | 23.3 | 47.9 | 52.4 | 53.3 |
| Mistral 24B fine-tuned | 25.0 | 29.9 | 30.0 | 50.3 | 60.2 | 58.8 |

Table 4: Performance comparison across different models and evaluation settings. Task SR represents task success rate, while Constraint SR represents constraint satisfaction rate. The +URL and +History columns indicate additional information provided during evaluation.

**Dataset curation.** We consider the following variations in the choice of data for training: only successful trajectories, all trajectories (successful + unsuccessful), and partially successful trajectories. We find that learning from partially successful trajectories is better than learning from successful complete trajectories alone, which demonstrates that we benefit from scale. We also consider a multi-stage training strategy where we train on all trajectories, followed by successful trajectories only. While this approach leverages unsuccessful trajectories and benefits from data scale, it is inferior to the simpler single-stage training strategy with partially successful trajectories alone.

**Training Strategy.** We find that LoRA training is more effective than full finetuning. This shows that LoRA is more effective in unlocking the knowledge of the base PLM, while full finetuning is prone to overfitting, especially in our data regime. Overall, we find the simple training recipe of training relatively small LLMs with LoRA on partially successful trajectories to be an efficient and effective training recipe which significantly outperforms a much larger prompted 405B model.

### 3.3 Reliability of Automatic Evaluation

We study the reliability of our evaluation strategy by considering the inclusion/exclusion of information provided to the evaluator. We consider page screenshot as the minimum source of information about the agent state available to the evaluator and study the impact of including/excluding information such as the page URL and the history of actions performed by the agent (Table 4). First, we find that inclusion of URL leads to better evaluation scores as it can encode useful information such as location, dates, etc. that can be complementary to webpage contents. Consider the following example url: `https://www.momondo.com/security/check?out=%2Fcars%2FSan-Francisco%2CCA-c13852%2F2025-01-20%2F2025-01-20-13h`. In this case, even though the agent was eventually blocked by a captcha, it clearly made progress towards the task by setting the correct rental type, location and dates, as is evident from the url.

| Website | Wilbur (Lutz et al., 2024) | WebVoyager (Claude) (He et al., 2024) | WebVoyager (GPT-4o) (He et al., 2024) | WebVoyager (GPT-4V) (He et al., 2024) | Ours |
|---|---|---|---|---|---|
| Allrecipes | $60.0_{\pm0.0}$ | $45.9_{\pm3.4}$ | $56.3_{\pm1.3}$ | $51.1_{\pm2.2}$ | $\mathbf{62.5}_{\pm0.0}$ |
| Amazon | $43.9_{\pm0.0}$ | $58.6_{\pm4.2}$ | $53.7_{\pm2.5}$ | $52.9_{\pm1.4}$ | $\mathbf{63.2}_{\pm0.0}$ |
| Apple | $60.5_{\pm0.0}$ | $58.1_{\pm4.0}$ | $56.6_{\pm1.3}$ | $62.8_{\pm2.3}$ | $\mathbf{63.6}_{\pm0.0}$ |
| ArXiv | $51.2_{\pm0.0}$ | $55.0_{\pm7.0}$ | $\mathbf{60.5}_{\pm0.0}$ | $52.0_{\pm1.3}$ | $59.5_{\pm0.0}$ |
| GitHub | $22.0_{\pm0.0}$ | $56.9_{\pm1.4}$ | $57.7_{\pm3.7}$ | $59.3_{\pm3.7}$ | $\mathbf{62.5}_{\pm0.0}$ |
| Booking | $38.6_{\pm0.0}$ | $19.0_{\pm1.3}$ | $43.9_{\pm3.5}$ | $32.6_{\pm2.7}$ | $\mathbf{55.0}_{\pm0.0}$ |
| ESPN | $\mathbf{59.1}_{\pm0.0}$ | $46.2_{\pm1.3}$ | $44.0_{\pm2.7}$ | $47.0_{\pm1.3}$ | $47.5_{\pm0.0}$ |
| Coursera | $51.1_{\pm0.0}$ | $68.2_{\pm1.3}$ | $65.1_{\pm2.8}$ | $57.9_{\pm2.7}$ | $\mathbf{70.0}_{\pm0.0}$ |
| Cambridge Dict. | $\mathbf{86.0}_{\pm0.0}$ | $71.3_{\pm3.6}$ | $82.2_{\pm1.3}$ | $71.3_{\pm1.3}$ | $76.7_{\pm0.0}$ |
| BBC News | $\mathbf{81.0}_{\pm0.0}$ | $66.7_{\pm4.8}$ | $54.8_{\pm2.4}$ | $60.3_{\pm2.8}$ | $77.1_{\pm0.0}$ |
| Google Flights | $0.0_{\pm0.0}$ | $15.1_{\pm5.5}$ | $28.6_{\pm0.0}$ | $51.6_{\pm1.4}$ | $\mathbf{66.7}_{\pm0.0}$ |
| Google Map | $39.0_{\pm0.0}$ | $55.3_{\pm1.4}$ | $56.9_{\pm2.8}$ | $\mathbf{64.3}_{\pm2.8}$ | $60.5_{\pm0.0}$ |
| Google Search | $67.4_{\pm0.0}$ | $72.9_{\pm1.3}$ | $63.6_{\pm1.3}$ | $\mathbf{77.5}_{\pm2.7}$ | $75.0_{\pm0.0}$ |
| Huggingface | $53.5_{\pm0.0}$ | $53.5_{\pm4.7}$ | $42.6_{\pm3.6}$ | $55.8_{\pm2.3}$ | $\mathbf{68.6}_{\pm0.0}$ |
| Wolfram Alpha | $\mathbf{65.2}_{\pm0.0}$ | $51.5_{\pm5.4}$ | $\mathbf{65.2}_{\pm2.2}$ | $60.9_{\pm2.2}$ | $58.7_{\pm0.0}$ |
| Overall | $52.6_{\pm0.0}$ | $52.8_{\pm1.4}$ | $55.5_{\pm0.8}$ | $57.1_{\pm0.2}$ | $\mathbf{64.5}_{\pm0.0}$ |

Table 5: **Comparison of various models on WebVoyager.** For each automatic evaluation, we run GPT evaluator three times to calculate the performance mean and standard deviation.

On the other hand, we notice that inclusion of action history leads to generous evaluation due to hallucination of the judge model. For instance, if the action 'type [3] [New York]' is provided in the agent history, the judge often simply assumes that the action of typing 'New York' into the relevant field in the page was successful, regardless of whether the action was successfully executed in the page. We further confirmed that inclusion of action history leads to evaluation outputs the correlate less with human evaluation. These ablations indicate that url and page contents provide the most unbiased signal for evaluation.

### 3.4 WebVoyager Evaluation

**Baselines.** We consider Wilbur (Lutz et al., 2024) and the agents reported in WebVoyager (He et al., 2024) as baselines. For fair comparison, we only compare with baselines that evaluate performance based on the standard evaluation protocol proposed in WebVoyager [1]. We use GPT-4o for evaluation since the GPT-4V model used in these works is deprecated.

**Evaluation Protocol.** We follow the standard evaluation protocol proposed in He et al. (2024). The final 15 screenshots of the agent trajectory, along with any final textual responses are provided to the VLM-judge which is required to output a success/failure judgement. Performance average and standard deviation are calculated based on three runs of the GPT-4o evaluator, similar to He et al. (2024). Note that this evaluation protocol is different from the evaluation metrics considered in previous sections – we only consider this evaluation approach for this dataset for consistency with prior reported results.

**Results.** We compare our approach against prior work in Table 5. We outperform prior approaches based on large closed-source foundation models such as GPT and Claude with a 24B parameter small language model. In particular, we achieve significant performance improvements on hard domains such as Booking and Google Flights, where we achieve improvements of 11.1% and 15.1% respectively, compared to the best baseline approaches on those domains. To our knowledge, our work is the first to achieve competitive performance

---

[1]Agent-E (Abuelsaad et al., 2024) and Browser Use (Müller & Žunič, 2024) used human evaluation, and other commercial UI control systems did not reveal their evaluation setting.

on the WebVoyager benchmark with a fine-tuned small language model. We also note that the all standard deviations are less than 0.05 due to high self-agreement of GPT-4o.

## 4 Related Work

**Benchmarks.** Evaluating web navigation agents presents significant challenges due to complex observations, multi-step interactions, and incomplete information. To address these challenges, prior work have proposed static benchmarks such as Mind2Web (Deng et al., 2023) and WebLINX (Lù et al., 2024), which provide efficient evaluation frameworks with step-level assessment by comparing predicted actions against reference ground-truth actions. However, results from these benchmarks may not accurately reflect agent performance on real-world tasks. Alternatively, benchmarks using live websites, including WebShop (Yao et al., 2022), WebArena (Zhou et al., 2023), and VisualWebArena (Koh et al., 2024), enable controlled evaluation on fixed, simulated websites but often fail to capture the complexity, stochasticity, and dynamic nature inherent in real-world web environments. WebVoyager (He et al., 2024) addresses this limitation by proposing evaluation tasks and automatic evaluators for task execution on actual websites, making it one of our chosen evaluation benchmarks. To further assess web agents on highly complex tasks, we introduce our own evaluation benchmark, BookingArena, which comprises real-world booking tasks that better reflect the challenges agents face in practical applications.

**Automatic Evaluation for Web Agents.** Existing web agent evaluation approaches face significant limitations in both scalability and reliability. WebCanvas (Pan et al., 2024b) and VisualWebArena (Koh et al., 2024) rely on human-annotated subgoal states and symbolic reward functions respectively, which lack scalability for broader applications. Conversely, WebVoyager (He et al., 2024) and other approaches (Pan et al., 2024a) employ VLM/LLM judges that evaluate success based on final screenshots and agent responses, but these methods prove unreliable and imprecise for complex, multi-criteria tasks. To address these challenges, we propose a constraint-based automatic evaluation framework for structured tasks defined by specific constraint sets, offering a more systematic approach than existing VLM/LLM-based methods while maintaining greater scalability than human-specified reward functions.

**Automatic Trajectory Data Generation for Web Agents.** Automatic generation of trajectory data for training web agents remains a relatively underexplored area. InSTA (Trabucco et al., 2025) uses LLMs to generate web tasks and evaluate agent-generated trajectories, while Explorer (Pahuja et al., 2025) prompts LLMs to propose and refine tasks, performing screenshot-augmented evaluation of agent trajectories. Although we share the scaling focus with these works, their evaluation frameworks primarily rely on the heuristic capabilities of evaluator models to assess trajectories based on natural language requirements, which limits their effectiveness for complex tasks. In contrast, we introduce a constraint-based framework for fine-grained evaluation, enabling our pipeline to collect high-quality trajectories for challenging tasks.

## 5 Conclusion

In conclusion, we presented a scalable, automatic data generation and distillation pipeline for developing proficient small language model agents. We introduce a novel automatic evaluation approach for reliably assessing agent trajectories. Using these metrics, we demonstrate how partial trajectories can be effectively leveraged to scale up high-quality training data. Our trained small language model agent, with 24B parameters, outperforms most prior approaches based on much larger closed-source foundation models, both on a new benchmark we propose and the Webvoyager benchmark. We also contribute a novel benchmark focused on challenging booking tasks, which we hope will be valuable in advancing the development of agents for complex web tasks. Directions for further improvement of our work include extending support to multimodal observations and broadening the scope to other platforms and environments beyond the web.

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

# A   Appendix

```
I am trying to generate examples of realistic tasks that our customers might ask their AI assistants to complete
on the internet.

You must create tasks that adhere the specific categories of tasks as expressed by these examples:
{examples}

Each task instruction should fulfill the following criteria:
(1) Precision: It should be sufficiently precise for the AI assistant to complete the task without having to ask
followup questions.
(2) Diversity: The tasks should be diverse. I don't want a dataset of tasks that are all the same. For instance,
DO NOT make every task related to adding an item to a shopping cart (at most, cart-related tasks should be 50%
of the tasks) or searching using a search bar.
(3) Realism: The tasks should be realistic. Think about what our human customer base might actually try to
accomplish using our AI assistant.
(4) Complexity: The task should be very simple for easy difficulty, then scale up for harder difficulties by
adding additional constraints to the task.
(5) Free: The AI assistant should not have to spend money in order to complete the task. For instance, if the
task regards a potential purchase, it should always conclude by adding it to the cart or identifying the booking
rather than actually purchasing it. The AI assistant has no access to payment methods (e.g., credit cards).
(6) Credential-Free: The AI assistant should not need access to our customers' personal data to complete the
task. So, for instance, filling out a form that requires a user's name or address, or logging into a real online
account, is off limits.
(7) Trajectory-Centric: Tasks should not require the AI assistant to report back on results. We have a separate
AI model that extracts the correct information from the webpages that are visited. Thus, the task should only
require the assistant to perform the correct actions on the web necessary to yield the correct webpage (e.g.,
visiting a webpage containing the correct information) or to elicit the right results (e.g., adding it to the
cart) from a website.
(8) High-Level: Tasks should not include low-level details (e.g., "filter to show only refrigerators with water
dispensers") because it is difficult to know what GUI interactions or filters might be available on a particular
website. Instead, tasks should be high-level, expressing a goal that can be accomplished by methods determined
by the AI assistant.
(9) Dated: For any task involving a booking date, the date should be between one to four months after
{current_date}. Tasks should always be expressed using that format (i.e., Month DD, YYYY such as "December 17,
2025")
(10) Relevance: Focus on the core tasks available on each website using your specific knowledge of that site and
the functionalities available on it, such as UI features and filering and search options.

Your job will be to generate {num} task instructions for EACH difficulty level for the website {website}. You
have already generated the following list of task instructions for this website. Please make sure that your new
task instructions are unique (i.e., they should be different from this list).
{prev_task_instructions}

Based on the above criteria, and using the examples as inspiration, please generate {num} task instructions for
each difficulty level for the website {website}. Please separate each task by a single carriage return ONLY.
```

Figure 3: Task generation prompt template used to generate diverse tasks for each website. The template includes specific criteria for task generation and ensures consistency across different difficulty levels while maintaining website-specific rules.

```
You are an autonomous intelligent agent tasked with navigating a web browser. You will be given web-based tasks.
These tasks will be accomplished through the use of specific actions you can issue.

Here's the information you'll have:
The user's objective: This is the task you're trying to complete.
The current web page's accessibility tree: This is a simplified representation of the webpage, providing key
information. It enumerates all the elements in the current web page in the format [id] [tagType] [text content]
[(optional) properties] where tagType is the type of the element (e.g., button, link), text content is the text
content of the element, and properties is an optional field which lists the properties of the element. For
example, [1234] [button] [Add to Cart] means that there is a button with id 1234 and text content 'Add to Cart'
on the current web page.
The current web page's URL: This is the page you're currently navigating.
The open tabs: These are the tabs you have open.
The previous reasonings: These are the past reasonings for taking corresponding actions.
The previous action: These are the actions you have performed. It may be helpful to track your progress.

The actions you can perform fall into several categories:

Page Operation Actions:
'''click [id]''': This action clicks on an element with a specific id on the webpage.
'''type [id] [content]''': Use this to type the content into the field with id. By default, the "Enter" key is
pressed after typing unless press_enter_after is set to 0, i.e., '''type [id] [content] [0]'''.
'''hover [id]''': Hover over an element with id.

URL Navigation Actions:
'''goto [url]''': Navigate to a specific URL.
'''go_back''': Navigate to the previously viewed page.
'''go_forward''': Navigate to the next page (if a previous 'go_back' action was performed).

Completion Action:
'''stop [answer]''': Issue this action when you believe the task is complete. If the objective is to find a
text-based answer, provide the answer in the bracket.

To be successful, it is very important to follow the following rules:
1. You should only issue an action that is valid given the current observation
2. You should only issue one action at a time.
3. You should follow the examples to reason step by step and then issue the next action.
4. Generate the action in the correct format. Start with a "In summary, the next action I will perform is"
phrase, followed by action inside '''''''. For example, "In summary, the next action I will perform is '''click
[1234]'''".
5. Issue stop action when you think you have achieved the objective. Don't generate anything after stop.
6. Note that there is no need to scroll since the entire current page is given.

OBSERVATION:
[1744] [link] [HP CB782A#ABA 640 Inkjet Fax Machine (Renewed)]
[1749] [StaticText] [$279.49]
[1757] [button] [Add to Cart]
[1760] [button] [Add to Wish List]
[1761] [button] [Add to Compare]
URL: http://onestopmarket.com/office-products/office-electronics.html
OBJECTIVE: What is the price of HP Inkjet Fax Machine?
PREVIOUS REASONINGS: None
PREVIOUS ACTION: None

Let's think step-by-step. This page list the information of HP Inkjet Fax Machine, which is the product
identified in the objective. Its price is $279.49. I think I have achieved the objective. I will issue the stop
action with the answer. In summary, the next action I will perform is ```stop [$279.49]'''

...
```

Figure 4: Prompt used for few-shot agent.

```
You are an expert in assessing the performance of a web navigation agent, whose role is to help a human user
navigate a website to complete a task. You will be given the following information:

* The user's intent: This is the task the agent is trying to complete
* Constraints: These are the constraints the agent needs to fulfill in order to complete this task
* The screenshot of the final webpage of the agent's task execution
* The URL of the final webpage of the agent's task execution

Based on this information, your task is to evaluate the agent's trajectory based on the following guidelines.
Please reason step by step before giving the final judgement. Take a close look at the screenshot of the final
webpage and the URL to see whether it meets the user's requirement.
* For each constraint that given, use the process described below to check whether the constraint was met by the
agent; to be considered met, the correct value of constraint must be either explicitly present on the page or
the URL.
* Note that the constraint 'made_selection' corresponds to locating one specific listing. This constraint is
unmet if the agent is on a search results page at the end; for it to be met, the agent MUST be on a page
dedicated to one specific listing.
* If the value of a particular constraint cannot be determined based on the page screenshot, check the URL to
see if the information is present.

Output your evaluation in valid json format as shown in the example below, where, for each constraint:
* "ground_truth" represents the the value of the constraint given to you
* "agent_state" is the observed value of the constraint on the agent's final webpage
* "matching" represents whether the constraint is met (true/false) in the agent's final webpage, based on
whether agent_state matches ground_truth

{
    "made_selection": {
        "ground_truth": true,
        "agent_state": "true",
        "matching": "true",
    },
    "location": {
        "ground_truth": "Aspen, CO",
        "agent_state": "Aspen",
        "matching": "true",
    },
    "unit_type": {
        "ground_truth": "vacation rental",
        "agent_state": "retreat house",
        "matching": "true",
    },
    "start_date": {
        "ground_truth": "January 08, 2025",
        "agent_state": "January 11, 2025",
        "matching": "false",
    },
    "end_date": {
        "ground_truth": "January 12, 2025",
        "agent_state": "January 12, 2025",
        "matching": "true",
    }
}
```

Figure 5: Prompt used for constraint evaluation with screenshot observations and VLM judge.

```
You are an expert in assessing the performance of a web navigation agent, whose role is to help a human user
navigate a website to complete a task. You will be given the following information:

* The user's intent: This is the task the agent is trying to complete
* Constraints: These are the constraints the agent needs to fulfill in order to complete this task
* The accessibility tree of the final webpage of the agent's task execution
* The URL of the final webpage of the agent's task execution

Based on this information, your task is to evaluate the agent's trajectory based on the following guidelines.
Please reason step by step before giving the final judgement. Take a close look at the accessibility tree of the
final webpage and the URL to see whether it meets the user's requirement.
* For each constraint that given, use the process described below to check whether the constraint was met by the
agent; to be considered met, the correct value of constraint must be either explicitly present on the page or
the URL.
* Note that the constraint 'made_selection' corresponds to locating one specific listing. This constraint is
unmet if the agent is on a search results page at the end; for it to be met, the agent MUST be on a page
dedicated to one specific listing.
* If the value of a particular constraint cannot be determined based on the page accessibility tree, check the
URL to see if the information is present.

Output your evaluation in valid json format as shown in the example below, where, for each constraint:
* "ground_truth" represents the the value of the constraint given to you
* "agent_state" is the observed value of the constraint on the agent's final webpage
* "matching" represents whether the constraint is met (true/false) in the agent's final webpage, based on
whether agent_state matches ground_truth

{
    "made_selection": {
        "ground_truth": true,
        "agent_state": "true",
        "matching": "true",
    },
    "location": {
        "ground_truth": "Aspen, CO",
        "agent_state": "Aspen",
        "matching": "true",
    },
    "unit_type": {
        "ground_truth": "vacation rental",
        "agent_state": "retreat house",
        "matching": "true",
    },
    "start_date": {
        "ground_truth": "January 08, 2025",
        "agent_state": "January 11, 2025",
        "matching": "false",
    },
    "end_date": {
        "ground_truth": "January 12, 2025",
        "agent_state": "January 12, 2025",
        "matching": "true",
    }
}
```

Figure 6: Prompt used for constraint evaluation with accessibility tree text observations and LLM judge.

```
You are an expert in assessing the performance of a web navigation agent, whose role is to help a human user
navigate a website to complete a task. You will be given the following information:
* The user's intent: This is the task the agent is trying to complete

* For the given intent, identify the key constraints mentioned in the intent. For example, the intent 'On
vacasa.com, find a vacation rental in Aspen, CO for January 08, 2025, to January 12, 2025, that accommodates 8
guests and has ski-in/ski-out access.' mentions the following constraints: made_selection, location, rental
type, start date, end date, number of guests, ski-in/ski-out access.
* If the intent involves locating one specific listing, and the intent does not use the word search, then create
an addition constraint called 'made_selection'. This constraint is unmet if the agent is on a search results
page at the end; for it to be met, the agent MUST be on a page dedicated to one specific listing. You must
abolutely not forget to include the made_selection constraint in these cases. When in doubt, include it.

Here's an example.

OBJECTIVE: Locate an apartment in Tokyo, Japan, available on January 30, 2025.
CONSTRAINTS:
- made_selection: True
- location: Tokyo, Japan
- date: January 30, 2025
- locate_apartment: true
```

Figure 7: Prompt used for constraint extraction.

```
You are an intelligent web agent. Given the following information, your task is to predict the next action.
Task: This is the task the agent is trying to complete.
Previous Actions: The previous actions the agent has performed.
URL: The URL of the current webpage
Page accessibility tree: The accessibility tree representation of the current webpage.

Task: <task>
Previous Actions: <past_actions>
URL: <url>
Page accessibility tree: <page_text>
```

Figure 8: Prompt used for finetuning.

```
https://www.trivago.com
https://www.homeaway.com
https://www.google.com/flights
https://www.vrbo.com
https://www.rome2rio.com
https://www.orbitz.com
https://www.hilton.com
https://www.booking.com
https://www.airbnb.com
https://www.travelocity.com
https://www.plumguide.com
https://www.ihg.com
https://www.hotels.com
https://www.vacasa.com
https://www.sonder.com
https://www.redawning.com
https://www.radissonhotels.com
https://www.momondo.com
https://www.megabus.com
https://www.kayak.com
```

Figure 9: Websites used for evaluation.

