# OpenReview forum: "Scaling Web Agent Training through Automatic Data Generation and Fine-grained Evaluation"
_colmweb.org/COLM/2025/Conference — COLM 2025_

### Official Review · Reviewer_3xdz · 2025-04-27

**Rating:** 7
**Confidence:** 4
**Ethics Flag:** 1

**Summary:**

The paper proposes a novel training strategy for fine-tuning smaller language models to be used as browser (web) agents. In particular, they propose an automatic data collection strategy using a novel evaluation regime based on assessing trajectory over multiple independent constraints, automatically extracted from the agent's goal. The automatically obtained data is then used to fine-tune an LLM, achieving performance competitive or above that of significantly larger models, without additional manually annotated training data.

**Questions To Authors:**

1. I don't fully understand the experiment in Section 3.3. Is there a ground truth value for the evaluator? Why should we assume that an evaluator that assess the underlying model as correct more often is itself more correct than a harsher one?

2. Related, I think the text in lines 257-267 should be in Section 3.4

3. Did you consider using the same model to both generate trajectories and fine-tune? What would you expect in that case?

4. WebVoyager uses screenshots from the last 15 steps of the agent for evaluation, while BookingArena only uses the final screenshot (line 201). Why?

5. The prompt in Fig. 7 is booking specific. What did you use for other WebVoyager tasks?

6. It seems the paper is using a different font than the COLM template. Not sure if this was intentional.

**Reasons To Accept:**

While the idea of decomposing LLM judges into constraints is not new (see for example the survey by Gu et al at https://arxiv.org/abs/2411.15594), it's application to web agents is both interesting and non-trivial, as it is made possible by the novel automatic constraint decomposition from the agent goal.
Additionally, the use constraint decomposition seems very powerful, as it allows to use partial trajectories where the agent succeeded up to a point, yet failed the overall task. This process greatly increases the amount of training data that can be obtained even with a relatively weaker teacher. The paper further improves on this with the idea of retroactively adjusting the goal based on whatever partial trajectory the agent achieves.

Finally, the proposed benchmark can be useful to the community, as it uses very challenging tasks (booking tasks are among the hardest in WebVoyager) but it also considers more websites than tasks, allowing to measure generality across websites.

**Reasons To Reject:**

My main concern with the paper is that the method relies on decomposing the goal into independent constraints.
While this seems plausible in search and transaction tasks such as booking tasks, it is less clear how possible this is in general, especially with goals that do not provide obviously extractable slot values, that provide soft constraints, or that require alternative reasoning ("if X is possible I want X", "either X or Y", etc). These situations do not appear in WebVoyager, and I would love to know if these were explored or considered in designing BookingArena. If not, additional evaluation beyond booking tasks might be necessary.

Related, there is very little discussion or evaluation of the constraint extraction itself. It is not clear if the method relies on defining some sort of task-specific ontology (slot names), or the LLM can infer reasonable constraints itself. It would also be very helpful to see whether the LLM can extract accurate constraints, especially across different tasks in WebVoyager.

Secondarily, the use of partial trajectories seem to provide only a small boost in performance (Table 2), despite a much larger training set. I would love to see additional ablations investigating which partial trajectories are useful, and also specifically whether the idea of retroactively rewriting goals to match the agent trajectory has any measurable impact.

---

> ### Author Response · Authors · 2025-06-03
> **Author Response**
>
> We thank the reviewers for their insightful comments. Reviewers noted that we present an effective approach for automatic evaluation and partial trajectory extraction using constraints (puzT, mfjm, 3xdz, 3mSW), demonstrate significant improvements over baselines on key benchmarks (puzT, mfjm, 3mSW), and introduce a new benchmark valuable for future research (mfjm, 3xdz, 3mSW). We address specific review points below.
>
> **Ability to handle soft constraints**
> Our framework does not restrict the form of constraints -- In principle, they can capture arbitrarily complex constraints (such as if/else constraints). However, such scenarios do not appear in our benchmark.
> Our benchmark is a step towards studying agents in longer horizon and more structured tasks. As we demonstrate, existing agents such as Browser Use and Claude Computer Use which have been shown to perform well in other tasks struggle in this setting.
> We leave the exploration of more complex constraint types to future work.
>
> **Evaluation of constraint extraction**
> Note that we do not use any predefined ontology to derive the constraints and LLMs are used to generate the constraints as free form text. We leverage powerful LLMs (e.g., LLaMA 3, GPT) to extract constraints, resulting in highly accurate outputs. To evaluate their quality, we conducted a human study on 120 test tasks. Annotators were presented with a task description and asked to identify an exhaustive set of constraints. Our key findings were:
> - In all cases, the constraints identified by human annotators were a subset of those extracted by the LLM. In some instances, the same constraint was expressed differently (e.g., a human might annotate "trip type: one-way" while the LLM outputs "flight\_type: one-way"). We treat such semantically equivalent constraints as matches.
> - In a few cases, human annotators failed to capture all constraints accurately. For example, when a task required both departing and return flights to be non-stop, human annotator only provided "non-stop: True", which can be ambiguous.
>
> In summary, constraint extraction using LLMs proved to be highly accurate and robust for our test tasks.
>
> **use of partial trajectories seem to provide only a small boost in performance**
> When leveraging partial trajectories, task success rate improves from 16% to 25% compared to using successful trajectories only, which is a relative improvement of 56%.
>
> **Which partial trajectories are useful**
> Success/failure is a strict metric that can limit data yield, especially on harder websites with low success rates. Partial trajectories let us capture meaningful progress made by the teacher agent, even if the task wasn't fully completed (e.g., Figure 2). Including them is essential for scaling data coverage and learning from a wider range of agent behaviors.
>
> **Why should we assume that an evaluator that assess the underlying model as correct more often is itself more correct than a harsher one?**
> Our main intention was to point out that environment-based signals (e.g., screenshots, URLs) are more reliable than agent-based signals (e.g., action histories), which often lead to hallucinations.
> For instance, comparing screenshot only vs screenshot + action history, the former is more stringent, and in fact more reliable, as inclusion of action history leads to hallucination (which was also verified with human evaluation).
> However, when comparing screenshot only vs screenshot + url, although the former is more stringent, it also penalizes cases where agent made progress that is not reflected in the final page (we provide an example in line 246). The url can complement information in the screenshot, providing a more holistic view of the agent's progress.
> So our conclusion isn’t that lenient evaluators are better, but that evaluation should rely on observable environment signals, which are more trustworthy.
>
> **same model for generating trajectories and fine-tuning**
> This is an interesting case, but we haven't tried it. But we do expect improvements in this case as well.
>
> **Why use only final screenshot for evaluation?**
> Reasoning over multiple screenshots introduces complexities, such as detecting whether a previously satisfied constraint was later undone by another action. In contrast, the final page more directly reflects the cumulative outcome of all agent actions. This simplifies the evaluation process and enables the use of LLMs/VLMs that may not reliably reason over multiple observations.
>
> **WebVoyager evaluation**
> Note that we do not use constraint-based evaluation for reporting WebVoyager performance - Rather, we use the Webvoyager evaluation protocol .
>
> **Formatting**
> Thanks for pointing out the formatting issues, we will fix this in the revision.

---

> > ### Comment · Reviewer_3xdz · 2025-06-09
> >
> > Thank you for answering my review.
> > I'm not sure how your framework can actually handle soft-constraints, especially inconsistent constraints that cannot be satisfied at the same time, given that your main metric for keeping training data requires matching all constraints. I would love more elaboration on that in the final revision.
> >
> > Thank you also for providing results on the constraint extraction evaluation. Those are very interesting and show the strength of the approach.
> >
> > Overall, this is a good paper, and I'm keeping my score.

---

### Official Review · Reviewer_puzT · 2025-05-07

**Rating:** 6
**Confidence:** 4
**Ethics Flag:** 1

**Summary:**

This paper aims to build a high-quality data generation (more precisely, selection) pipeline to improve web agent training, by proposing fine-grained evaluations on the progress made towards task completion, instead of sparse binary rewards. Experiments on self-created BookingArena benchmark demonstrate the effectiveness of the proposed method in distilling better student models out of data synthesized by the teacher model.

**Questions To Authors:**

1. Wrong template? At least the font is off.

2. For the “Constraint Framework”, since it’s using LLM, how to make sure that the set of constraints is stable (for similar tasks ,the constraints are similar) and comprehensive (compared to humans to come up with the constraint, how does LLM perform)?

3. Based on CSR, do you only discard intermediate steps that fail? How about the following steps?

**Reasons To Accept:**

1. **Valid Solution for the Important Evaluation Issue**: This paper tackles a pressing issue for web agents -- automatic evaluation, by measuring a trajectory's progress towards success instead of a binary 0/1 reward. The method (by breaking the tasks down into multiple constraints) is reasonable, and if producing reasonably valid scores, can be utilized to better train agent policy/reward models.

2. **Effectiveness of the Proposed Method** Experiment results demonstrate the big improvements brought by the proposed method, demonstrating its effectiveness in yielding better student agent models.

**Reasons To Reject:**

1. **Significance of Results?** As there are only 120 self-created tasks in the experimented BookingArena dataset (what makes it worse is that each website only has 6 tasks? which is far from a good coverage of different tasks and contexts), it is unclear if the performance differences between methods are statistically significant, and thus make it the validity of conclusions questionable.

2. **Reliance on Multi-Faceted Queries** This work creates special tasks that involve multiple constraints in travel booking scenarios, which is a relatively limited scenario, if considering the numerous websites and varied tasks one could do on the web. It is unclear if this method (particularly constraint-based evaluation) could be similarly applied to benefit (i) simple tasks, (ii) long-horizon tasks, e.g., with long procedures but not many constraints, or other common scenarios.

3. **Limitation: Performance Bounded by Teacher Model** The proposed method is essentially selecting high-quality data synthesized by an existing model, which means the to-be-trained student model is bounded by the teacher's performance. This method is more of a better data selection (for better model distillation), than advancing the upper-bound of top-performing agents.

---

> ### Author Response · Authors · 2025-06-03
> **Author Resopnse**
>
> We thank the reviewers for their insightful comments. Reviewers noted that we present an effective approach for automatic evaluation and partial trajectory extraction using constraints (puzT, mfjm, 3xdz, 3mSW), demonstrate significant improvements over baselines on key benchmarks (puzT, mfjm, 3mSW), and introduce a new benchmark valuable for future research (mfjm, 3xdz, 3mSW). We address specific review comments below.
>
> **Significance of Results**
> In order to evaluate agents on a larger set of tasks, we doubled the number of tasks per website. Due to limited time, we compare our approach against Operator and UI-TARS, the top performing methods for this experiment (We will include Claude Computer Use and Browser Use results in the revision).
> We present the result on all 240 tasks below (20 websites, 12 tasks per website), along with the delta from the previous result (on 120 tasks). We find that performance is generally consistent with our previous runs. The most significant delta was observed for Operator SR, but this does not alter the relative performance trend among models.
>
> | Method     | SR            | CSR            |
> |------------|---------------|----------------|
> | UI-TARS    | 6.7% (+0.3%)  | 46% (+1%)      |
> | Operator   | 21.8% (+3.8%) | 63.5% (+1.5%)  |
> | Ours       | 25% (±0%)     | 55.5% (-0.5%)  |
>
> **"travel booking scenarios is limited"**
> While our work focuses on travel booking tasks, the approach naturally extends to other structured, constraint-driven domains such as e-commerce. These tasks are common in real-world applications and pose challenges due to their long-horizon nature. Notably, they are (i) underrepresented in existing benchmarks like WebVoyager and (ii) difficult for prior methods (e.g., Browser Use, Claude Computer Use) to solve. For these reasons, we believe our booking benchmark offers a valuable contribution toward advancing the capabilities of autonomous agents.
>
> **applicability to tasks not defined in terms of explicit constraints**
> Due to the fuzzy nature of the constraints, our approach is also applicable to tasks that are not defined in terms of explicit constraints. For instance, we show that our approach works well on search tasks from the WebVoyager benchmark.
>
> **Performance Bounded by Teacher Model**
> We note that our fine-tuned student agent significantly outperforms the teacher model, despite also being much smaller. We present this comparison in Table 2, where the LLAMA 405B teacher agent achieves (SR, CSR) = (0.17, 0.48), whereas the Mistral 24B model trained on partial trajectories with LoRA achieves (SR, CSR) = (0.25, 0.56). We believe that large scale automated data generation coupled with effective data curation strategies can help advance the frontier of agents.
>
> **How do we ensure that the set of constraints is both stable and comprehensive?**
> We use powerful LLMs (e.g., LLaMA 3, GPT) to extract constraints as free-form text, resulting in high-quality outputs. To assess their accuracy, we conducted a human study on 120 test tasks. Annotators were given task descriptions and asked to exhaustively list all relevant constraints. Our key findings were:
> - In all cases, the human-identified constraints were a subset of those extracted by the LLM. In some cases, constraints were phrased differently but were semantically equivalent (e.g., “trip type: one-way” vs. “flight_type: one-way”), which we treat as matches.
> - In a few instances, annotators missed constraints. For example, when a task required both outbound and return flights to be non-stop, a human annotator provided only “non-stop: True,” which is ambiguous.
>
> In summary, constraint extraction using LLMs was found to be both accurate and robust across our test set.
>
> **Based on CSR, do you only discard intermediate steps that fail? How about the following steps?**
> As described in lines 133–136, we extract the trajectory prefix until timestep t' where t' is the timestep in the trajectory for which the most constraints were satisfied. The following steps (i.e., t > t') are discarded.
>
> **Formatting issues**.
> Thank you for pointing this out. We will fix formatting issues in the revision.

---

> > ### Comment · Reviewer_puzT · 2025-06-09
> >
> > The response answers most of my questions (for which I slightly increased my score)! However, for concerns on "significance of results", it might be better to perform significance testing, or computing result variance on multiple runs with the *same* number of examples.

---

### Official Review · Reviewer_3mSW · 2025-05-12

**Rating:** 7
**Confidence:** 4
**Ethics Flag:** 1

**Summary:**

This paper introduces a pipeline for training web agents by automatically generating and curating high-quality training data. Though certain parts of the method, such as generating tasks using LLMs and exploration, have been investigated in prior works, the proposed constraint-based evaluation framework to filter out low-quality trajectories and also provide a way of keeping partially successful trajectories is effective and interesting. CSR, the metric used for filtering trajectories, is also extendable for downstream evaluation. The paper also proposes a new benchmark, BookingArena, featuring complex booking tasks on 20 popular websites, and demonstrates that a relatively small model with only 24B parameters, when distilled, outperforms existing open-source models and matches or exceeds the performance of closed-source models. The approach is also validated on WebVoyager.

**Questions To Authors:**

1. Can you provide a list of example tasks generated by the LLM for generating the training data (the prompt in Fig 12 also lacks concrete few-shot examples but only mentions them while discussing the constraints) and also the distribution of the final generated dataset (300k action/obs) across the 1000 websites used? I am curious to see the intersection with booking websites in the training data used for the student models.
2. Can you provide more details about the task creation in BookingArena? It seems that the content is missing crucial details in this regard.

**Reasons To Accept:**

1. The constraint-based CSR metric is effective in filtering out trajectories for training using web agent data and helps in significantly increasing the size of the dataset. It is also a significant improvement over binary success/failure metrics generally used for evaluation, providing more granular insights into model performance that can also be used to improve them further.
2. The proposed method obtains competitive performance when compared to both open-source and closed-source models on BookingArena and WebVoyager.
3.  The BookingArena benchmark is another valuable contribution for diversifying web agent evaluation.

**Reasons To Reject:**

1. I believe that there are numerous steps where the pipeline relies on LLMs. I find it necessary to conduct a corresponding human study (even on a subset) to gauge the agreement of the proposed approach with human annotations.
2. While the method is described as scalable, the paper does not provide a detailed analysis of the computational cost associated with generating and evaluating large-scale trajectories. Per-step evaluation requires separate LLM/VLM calls that add up depending on the complexity of the task generated.

---

> ### Author Response · Authors · 2025-06-03
> **Author Response**
>
> We thank the reviewers for their insightful comments. Reviewers noted that we present an effective approach for automatic evaluation and partial trajectory extraction using constraints (puzT, mfjm, 3xdz, 3mSW), demonstrate significant improvements over baselines on key benchmarks (puzT, mfjm, 3mSW), and introduce a new benchmark valuable for future research (mfjm, 3xdz, 3mSW). We address specific review points below.
>
> **Computational cost**
> All experiments were conducted on two machines, each with 16 × A100 40GB GPUs.
> We deploy LLaMA 70B and LLaMA 405B (4-bit) models as LLM servers, each requiring 8 GPUs — allowing us to run up to four servers in parallel. These yield approximately 84k and 16k trajectories per week, respectively.
> For evaluation, we use 4-bit quantized LLaMA 70B models, which can fit in 2 A100 GPUs. Unlike trajectory generation, evaluation does not require web interaction and is highly parallelizable. The full dataset was evaluated with step-level evaluation in 4 days.
> We will include these computational details in the revision.
>
> **Booking Arena task creation**
> The prompt used for generating tasks in BookingArena is given in Figure 3. For each of the 20 websites, we used this prompt to generate 6 tasks covering all three difficulty levels (2 easy, 2 medium, 2 hard). We provide examples of easy, medium and hard tasks below.
>
> Example easy task: On booking.com, find a hotel available in Miami, FL for a single night on June 15, 2025.
> Example medium task: On ihg.com, locate a pet-friendly hotel in Orlando, Florida with free Wi-Fi, available for the dates June 17, 2025 to June 22, 2025.
> Example hard task: On kayak.com, find a multi-city flight itinerary starting from Chicago (ORD) to Paris (CDG) on June 10, 2025, and then from Paris (CDG) to Rome (FCO) on June 20, 2025, all in business class.
>
> **Human Studies**
> We conducted human studies to evaluate the quality of (i) constraint extraction and (ii) automatic evaluation. Our key findings are summarized below:
>
> * **Constraint Extraction**
>   Annotators were given 120 task descriptions from our test set and asked to exhaustively list all relevant constraints. In all cases, the human-identified constraints were a subset of those extracted by the LLM. In a few instances, annotators missed constraints. For example, when a task required both outbound and return flights to be non-stop, a human annotator provided only “non-stop: True,” which is ambiguous. Overall, constraint extraction using LLMs was found to be accurate and robust.
>
> * **Automatic Evaluation**
>   For each task, a human annotator was shown the final screenshot from the agent’s trajectory and asked to label each constraint as either *met* or *unmet*. We obtained the same judgments using an LLM. Agreement was defined as the fraction of constraints for which human and LLM judgments matched, and the overall score was computed by macro-averaging across tasks. Using this protocol, we observed an agreement score of 72.69% based on 75 evaluated trajectories.
>
> We will include further details about these studies in our revision.

---

> > ### Comment · Reviewer_3mSW · 2025-06-10
> >
> > Thank you for your responses! I would maintain my score. Good luck.

---

### Official Review · Reviewer_mfjm · 2025-05-12

**Rating:** 7
**Confidence:** 4
**Ethics Flag:** 1

**Summary:**

The paper proposes a method for synthesizing trajectories for training a web agent. Given a set of websites, prompted LLMs/VLMs are used to generate tasks, derive **target constraints** for the tasks (e.g. if the task asks to book a trip on April 8, one constraint would be “start_date: April 8”), sample trajectories by interacting with the websites, and score the trajectories against the constraints. The high-quality trajectories are used to LoRA-tune a Mistral 24B model.

Evaluation is done on a new benchmark BookingArena, which focuses on the travel booking task, and [WebVoyager](https://aclanthology.org/2024.acl-long.371/), which are mostly search tasks. The generated target constraints are used to compute partial credits during evaluation.

**Questions To Authors:**

1. What is the format of the BookingArena benchmark? Does it include local copies of the websites, or does it require interacting with actual websites. If actual websites are required, could the dataset become broken if the website changes?

2. Could the author give some statistics about the BookingArena benchmark? For example, do the tasks require navigating to different pages? How many required steps or target constraints are there on average? How diverse are the tasks?

3. Line 255 mentions human evaluation of the automatic evaluation method. Would it be possible to elaborate on this? What is the protocol, and what are the concrete statistics?

4. Why does the model fail completely on Radisson (Table 1) while the baselines get some partial rewards?

**Other suggestions and typos:**

* Lines 27-31: These are redundant with the previous paragraph.
* Lines 118-124: These could be compressed to something like “we average the scores” without having to write the full formulas.
* Line 244: “leads to leads to” → “leads to”
* Lines 257-267: These should probably be moved under Section 3.4.

**Reasons To Accept:**

1. **The software, training trajectories, and the BookingArena dataset** will be beneficial to other researchers. These are high-quality resources that would be cumbersome or expensive to generate.

2. **The proposed method is reasonable.** While the use of automatically generated constraints for validating model outputs is a known technique (e.g. in natural language generation), the application here is well-justified. It also allows partial credits based on intermediate steps in the trajectory that might not be present in the final task output (e.g. whether the model successfully clicked a page transition button does not appear in the final POST request).

3. The method empirically **outperforms reasonable baselines** (though see Weakness 1 below).

**Reasons To Reject:**

1. The comparison against the baselines might be **a bit unfair, as the proposed method is trained on a large amount of trajectories** (16k complete trajectories and 65k prefixes). This would be even more severe if **the test tasks have large overlap with the training tasks,** which seems to be the case if I am not missing something. The model would roughly know how to solve the task in the same distribution from the training trajectories. Granted, the closed-source baselines were probably also trained on a lot of web tasks (though in a different distribution), and the proposed method also works well on WebVoyager which is an existing dataset.

2. There are some parts that are missing details. Please see the “Questions To Authors” section below.

3. There is not a lot of **qualitative success and error analysis** of the models’ behaviors. For example, are there specific behaviors that the training trajectories help with? How well can the model generalize to tasks that are different from training tasks? The ablation runs in Table 2 could be a good source for the analysis

4. Using a bot to *intensively* interact with websites may **violate their terms of service.** Other web agent works try to sidestep this by working on local websites (e.g. [WebShop](https://proceedings.neurips.cc/paper_files/paper/2022/hash/82ad13ec01f9fe44c01cb91814fd7b8c-Abstract-Conference.html)) or, for companies, working with their product partners. This work scraped the top 1000 websites and even got blocked by some.

---

> ### Author Response · Authors · 2025-06-03
> **Author Response**
>
> We thank the reviewers for their insightful comments. Reviewers noted that we present an effective approach for automatic evaluation and partial trajectory extraction using constraints (puzT, mfjm, 3xdz, 3mSW), demonstrate significant improvements over baselines on key benchmarks (puzT, mfjm, 3mSW), and introduce a new benchmark valuable for future research (mfjm, 3xdz, 3mSW). We address specific review points below.
>
> **Overlap between train and test distributions**
> We examine the overlap between test tasks and training tasks.
> For each test task and website, there is no training task that has the same exact combination of constraints and values.
> When ignoring the values, the same exact combination of constraints appears in the training set for the same website for only 9/120 tasks. We further verify that, when excluding these 9 tasks from the evaluation, the task SR remains the same while CSR drops by 1%, indicating that these few overlapping cases are not driving the overall performance.
> In addition, we also demonstrate competitive performance on the WebVoyager benchmark, which shows the transferability of knowledge from our training tasks to unseen tasks.
>
> **Details about BookingArena benchmark**
> - Task format: Each task is identified by a starting url and task description. There are no local copies of the websites as this tends to limit the evaluation. Agents are required to interact with actual websites. The tasks are general enough that they will remain robust to website changes.
> - Number of steps: Yes, tasks often require multi-step navigation across different pages. The average length of successful trajectories from our fine-tuned agent in BookingBench is 10.6.
> - Number of target constraints: Each of the 20 websites in the test set have a total of 6 tasks (2 easy, 2 medium, 2 hard). We report the minimum, maximum and average number of constraints in the tasks in each difficulty category below.
> - Diversity: This shows that tasks are diverse both at the website level and difficulty level.
> We will incorporate these details into the revision.
>
> | Difficulty | Min | Max | Mean |
> |------------|-----|-----|------|
> | Easy       | 2   | 6   | 3.95 |
> | Medium     | 3   | 7   | 5.45 |
> | Hard       | 5   | 11  | 7.30 |
>
> **Error analysis**
>
> We provide difficulty-wise performance for our fine-tuned agent below. Performance naturally declines as the task difficulty and number of constraints increases.
>
> | Difficulty | SR   | CSR  |
> |------------|------|------|
> | Easy       | 0.43 | 0.68 |
> | Medium     | 0.21 | 0.56 |
> | Hard       | 0.13 | 0.45 |
>
> In contrast to LLM/VLM-as-judge approaches that produce a single success/failure judgment, our constraint-based evaluation offers fine-grained insights into which aspects of a task the agent struggles with. Specifically, we measure how often each constraint is successfully satisfied across tasks. Success rates for the 10 most frequent constraints are shown in the table below.
> Intuitively, location-related constraints have higher success rates, as they are often filled early in the task. In contrast, constraints related to amenities, such as pet-friendliness, are handled later in the task and have low success rates as a result, due to the dependency on prior actions.
>
> | Constraint           | Success Rate (%) |
> |----------------------|------------------|
> | location             | 79               |
> | departurelocation    | 73               |
> | destinationlocation  | 67               |
> | returndate           | 64               |
> | departuredate        | 56               |
> | startdate            | 55               |
> | petfriendly          | 50               |
> | enddate              | 47               |
> | madeselection        | 41               |
> | rentaltype           | 41               |
>
> **Human evaluation of automatic evaluation**
> Protocol: For each task, a human annotator was shown the final screenshot from the agent’s trajectory and asked to label each constraint as either met or unmet. We also obtained met/unmet judgments for each constraint using an LLM. Agreement score was defined as the fraction of constraints for which the human and LLM judgments matched. The overall agreement score was computed by macro-averaging across tasks.
>
> We evaluated 75 trajectories in this manner. When only the final screenshot was provided, the agreement score was 72.69%. When both the screenshot and the agent’s action history were provided, the agreement dropped to 69.42%. This suggests that including the agent’s actions and reasoning can sometimes mislead evaluators into believing certain constraints were satisfied, since agent actions may have failed or not achieved the intended outcome.

---

> > ### Author Response · Authors · 2025-06-03
> > **Author Response (Part 2)**
> >
> > **Are there specific behaviors the training trajectories help with?**
> > We include one qualitative example below to keep the response brief and will include additional examples in the revision.
> > In the booking.com website, the teacher agent often fails to realize that it has to click the 'Search' button after setting the location and date constraints for the search results to update. Instead, the teacher agent would directly click on a search result already on the current page, which would reset some of the constraints already set by the agent. Since our constraint based trajectory filtering approach will discard trajectory suffixes where the 'Search' button is not clicked (see Figure 2 of the paper for an illustration), the distilled agent learns the correct behavior from the valid trajectories.
> >
> > **Radisson failure mode**
> > On this particular website, the agent struggles to bypass the cookie popup. Instead of clicking 'Accept Cookies', it attempts to interact with other elements on the page—actions that fail due to the presence of the cookie banner. While the agent can handle cookie popups on other sites, it fails in this instance due to the specific positioning and implementation of the banner.
> >
> > **Websites terms of service**
> > Thank you for raising this important point. We acknowledge the ethical considerations with respect to accessing websites via automated means. We distributed data collection over a large number of websites to minimize server load and avoid causing disruptive behavior to any particular website. We chose to work with a public websites in order to study agent behavior in realistic and diverse environments, which is difficult to achieve on synthetic or partner-limited domains. That said, we recognize the trade-offs and will explore ways to mitigate this issue in future work.
> >
> > **suggestions and typos**
> > Thank you for these suggestions, we will incorporate these into our revision.

---

> > ### Comment · Reviewer_mfjm · 2025-06-04
> > **Thank you for the response.**
> >
> > **Overlap between train and test distributions:** The low task overlap is great. This should be highlighted in the paper.
> >
> > **Human evaluation of automatic evaluation:** The fact that human annotators are shown only the final screenshot is quite important. Some constraints can only be checked in the intermediate screenshots (e.g. the final screenshot may hide some payment information from the form in the previous page). The lower correlation could be because the human annotators didn't see the necessary information in the intermediate screenshot. I think this part needs further exploration.
> >
> > **Are there specific behaviors the training trajectories help with?:** This is an interesting example that would be great to showcase in the paper. I think this also addresses Reviewer puzT's Weakness 3: the student can perform well even when the teacher occasionally struggles, as long as there are *some* good trajectories from the teacher.
> >
> > Overall the response addresses my concerns (except for the human evaluation part) and I am increasing my score.

---

### Decision · Program_Chairs · 2025-07-08

**Decision:**

Accept

**Comment:**

The paper presents a constraint-driven pipeline that automatically generates and filters web-booking trajectories, then distills a 24B student that matches or outperforms a much larger teacher. All four reviewers leaned positive. They converge on strengths (clever constraint-based evaluation and a useful booking benchmark) and raise similar worries about evaluation rigor (train/test overlap, small task set, LLM reliance, compute). Author rebuttal helped clarify some of the overlap/compute doubts, but statistical-significance and “soft-constraints” remain partly open.

Strengths
1. The idea is novel. Fine-grained constraint evaluation can unlocks partial-trajectory reuse (mfjm, 3mSW, puzT, 3xdz)
2. The paper showed convincing performance (mfjm, 3xdz)
3. BookingArena benchmark will be a valuable resource (mfjm, 3mSW, 3xdz)
4. Reviewer mentioned that the authors had clear instructions and scripts in the opensource code (mfjm)

Weaknesses
1. Possible train–test leakage and baseline fairness (mfjm). The rebuttal provides overlap statistics) and additional ablations which addressed this.
2. Transparency about compute cost (3mSW). Rebuttal addressed by giving GPU counts & throughput
3. Limited statistical power on a 120-task benchmark (puzT) – authors doubled tasks but Reviewer suggested adding significance tests.
4. Reviewers were a bit concerned that the method is on narrow domains, with unclear transfer to simpler or “soft-constraint” tasks (puzT, 3xdz). Authors argue that method generalizes, but reviewers are still looking for concrete evidence beyond booking flows. They suggested some cross-domain experiments.
5. Reviewers asked about the reliance on LLM-extracted constraints (3mSW, 3xdz). Authors report a reasonable human-LLM agreement but reviewers encourages the authors to also add a discussion of failure modes and possible mitigations.

Overall, I think the paper makes valuable contributions, and would encourage the authors to make corresponding revisions as suggested by the reviewers.